# An Overview of the Design of Metal-Organic Frameworks-Based Fluorescent Chemosensors and Biosensors

**DOI:** 10.3390/bios12110928

**Published:** 2022-10-26

**Authors:** Ning Xia, Yong Chang, Qian Zhou, Shoujie Ding, Fengli Gao

**Affiliations:** College of Chemistry and Chemical Engineering, Anyang Normal University, Anyang 455000, China

**Keywords:** metal-organic frameworks, fluorescence, chemosensors, biosensors

## Abstract

Taking advantage of high porosity, large surface area, tunable nanostructures and ease of functionalization, metal-organic frameworks (MOFs) have been popularly applied in different fields, including adsorption and separation, heterogeneous catalysis, drug delivery, light harvesting, and chemical/biological sensing. The abundant active sites for specific recognition and adjustable optical and electrical characteristics allow for the design of various sensing platforms with MOFs as promising candidates. In this review, we systematically introduce the recent advancements of MOFs-based fluorescent chemosensors and biosensors, mainly focusing on the sensing mechanisms and analytes, including inorganic ions, small organic molecules and biomarkers (e.g., small biomolecules, nucleic acids, proteins, enzymes, and tumor cells). This review may provide valuable references for the development of novel MOFs-based sensing platforms to meet the requirements of environment monitoring and clinical diagnosis.

## 1. Introduction

Metal organic frameworks (MOFs), a class of porous-crystalline materials, are self-assembled from metal ions/clusters and organic ligands. They have been popularly applied in different fields, such as adsorption and separation, heterogeneous catalysis, drug delivery, light harvesting and chemical/biological sensing [1,2,3,4]. Various strategies have been proposed for the synthesis of MOFs, including hydrothermal/solvothermal reaction, solvent volatilization, and microwave-assisted, sonochemical and electrochemical methods. The influence of experimental parameters on the properties of MOFs have been investigated, such as metal ions, ligands, solvent, temperature, and reactant ratio. Many groups have summarized the progress in the advantages and disadvantages of different methods for the synthesis of MOFs [5,6,7,8,9,10]. Recently, MOFs have received special attention due to their versatile optical properties. Because some metal ions and organic ligands show adjustable fluorescence properties, a series of multi-dimensional MOFs with inherent fluorescence have been prepared and successfully implemented as fluorescent platforms for a wide range of applications, such as photocatalysis, chemical/biological sensing and biomedical imaging [11,12,13,14,15].

For the MOFs with intrinsic fluorescence, the emitting mechanisms can be classified into four subtypes: ligand-centered emission, metal ion-centered emission, ligand-to-metal charge transfer (LMTC)-based metal-centered emission, and metal-to-ligand charge transfer (MLCT)-based ligand-centered emission [16]. The resulting emission depends on the structure of MOFs, such as the spacing and orientation between linkers, the energy gap between linkers and metal units, the electronic configuration of metal ions and the bonding geometry. In addition, the aggregation-induced emission (AIE) luminogens (AIEgens) have been used as ligands to synthesize luminescent MOFs with AIE properties for various applications [17]. Moreover, the morphology and size of MOFs (e.g., spherical, wheat ear-like, and flower-like nanostructures) have a significant impact on the optical properties [18,19]. Fluorescent dyes and nanomaterials (e.g., quantum dots (QDs), carbon dots (CDs), and metal nanoclusters) can be integrated with MOFs as additional emission units by the ways of physical adsorption, covalent coupling and in-situ encapsulation [20,21,22,23]. For example, Rhodamine 6G (Rh6G) could be encapsulated into Zn-MOFs to form dual-emitting MOFs for the detection of 2,4,6-trinitrophenol (TNP) [24]. CdSe/ZnS core/shell QDs could be immobilized on porphyrin-based MOFs with the amine−Zn coordination interaction for enhancing light harvesting by energy transfer from QDs to MOFs [25]. Fluorescent CDs could be embedded into MOFs with molecularly imprinted polymers for the detection of quercetin [26]. Moreover, Ln^3+^ ions could bound with the functional groups of ligands in MOFs for emitting the characteristic fluorescence via the antenna effect [27].

During the past decades, MOFs-based fluorescent sensing platforms have been developed for the detection of various targets, including metal ions, small organic molecules and biomolecules [28,29,30,31,32]. Besides their unique optical properties, the high porosity and the abundant active sites of MOFs provide the spatial environment and multiple recognition sites and facilitate the adsorption of analyte in the pores, leading to the preconcentration of target and the enhancement of detection sensitivity. Recently, a few review papers have summarized the applications of MOFs-based materials in chemical/biological sensing [33,34,35,36,37,38,39]. For example, Wu’s group and Wang’s group reviewed the recent progress in fluorescence biosensors based on DNA–MOF hybrids [36,37]. Jia’s group discussed the post-synthetic modification methods for MOFs-based fluorescent sensors [38]. Li et al. summarized the achievements of CDs@MOFs-based sensors [39]. In this context, we systematically introduce the recent advancements of MOFs-based fluorescent chemosensors and biosensors, mainly focusing on the detection mechanisms and analytes. First, the common detection mechanisms in MOFs-based fluorescent sensing platforms are briefly introduced. Then, MOFs-based sensors for the detection of inorganic ions (i.e., metal ions cations and inions) are summarized. Next, the assays of small organic molecules using MOFs as fluorescence probes are reviewed. Last, most efforts were put into the summarization of recent advances in the field of MOFs-based sensing of biological species, such as small biomolecules, nucleic acids, enzymes, proteins and tumor cells. Additionally, the outlook and several issues of MOFs-based fluorescent sensing platforms are discussed. The aim of this review is to provide an extensive overview of a wide range of fluorescent MOF-based sensing methods.

## 2. Detection Mechanisms of MOFs-Based Fluorescent Sensing Platforms

Through a judicious selection of linkers and metal ions, functional groups, Lewis basic/acidic sites, and metal ions in MOFs can be used as the specific recognition sites for preferential responses toward targets. For example, MOFs with free-base porphyrin as the linker can bind to metal ions with high affinity, leading to the change of optical properties of porphyrinic MOFs [40]. Fluorescent MOFs containing pyridyl could be utilized for Cu^2+^ detection because of the strong affinity between pyridyl and Cu^2+^ ions [41]. Meanwhile, the quenching or increasing of the ligand fluorescence by extra metal ions can be used as a switch mode for target analysis. The general detection modes in fluorescence assays can be divided into three categories: signal-on, signal-off, and ratio analysis. MOFs can also be used as fluorescent sensing platforms to act as fluorescent probes or fluorescent modulators. The specific receptor–target interaction can induce the change in the fluorescence of MOFs via different mechanisms [42]. Herein, we briefly introduced four common detection mechanisms involved in most works, including photoelectron transfer (PET), Forster resonance energy transfer (FRET), inner filter effect (IFE), and target-induced chemical or structural change of MOFs (Figure 1) [43].

PET is an excited-state charge-transfer process in which the excited electron in the lowest unoccupied molecular orbital (LUMO) of the photo-excited donor can be transferred to the LUMO of acceptor, leading to the fluorescence quenching (Figure 1A) [44,45]. FRET with a distance-dependent non-radiative energy transfer process is the mostly common mechanism for fluorescence sensing. When the emission spectrum of donor partially overlaps with the absorption spectrum of acceptor, the energy is transferred from the donor to the acceptor through FRET process (Figure 1B) [46]. The FRET efficiency depends on several factors, including the spectral overlap extent, the distance between the donor and the acceptor, and the dipole–dipole interaction. Thus, when the excitation spectrum of targets overlaps with the emission spectrum of MOFs, the presence of targets (acceptors) can induce the change of the fluorescence of MOFs (donors). IFE involves the competitive absorption of the excitation light between analyte and MOFs or the adsorption of the fluorescent emission of MOFs, resulting in the obvious fluorescence quenching (Figure 1B) [46]. In this method, the adsorption spectrum of analyte always overlaps with the emission or adsorption of MOFs. Unlike PET and FRET, this process is independent on the distance between two species. Moreover, appropriate functional groups on ligands can influence the fluorescence properties of MOFs. They can act as the active sites for specific recognition of targets through coordination or covalent interaction, thus leading to the change of the fluorescence of MOFs. Furthermore, during the detection process, a target can induce the destruction of the skeleton of MOFs-based composites by interacting with the metal nodes or the ligands in MOFs, resulting in the change of fluorescence signal [47,48].

**Figure 1 biosensors-12-00928-f001:**
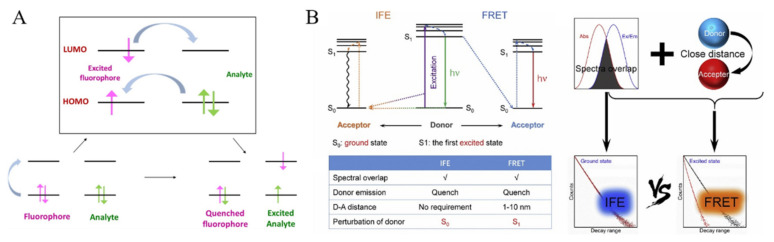
(**A**) Schematic illustration of the process of PET. Reproduced with permission [45]. Copyright 2020, Elesvier. (**B**) Schematic illustration of the process of IFE and FRET. Reproduced with permission [46]. Copyright 2019, Elsevier.

## 3. Chemosensors

Chemosensors based on the analyte-induced signal change are the most direct and convenient methods for sensing of ions, pollutants, pesticides, volatile organic compounds and drugs. Currently, a wide range of fluorescent chemosensors have been extensively exploited, including small organic molecules, semiconductor QDs, metal nanomaterials, CDs, and AIEgens-based chemosensors [49,50,51,52,53]. Compared to those fluorescent materials, MOFs exhibited several excellent advantages in the field of fluorescent assays. First, the tunable pore sizes of MOFs can act as the sieves to control the entry of analytes into the pores, resulting in the increased selectivity. Second, the large surface area of MOFs can adsorb and concentrate targets to high levels, leading to the enhanced sensitivity. Third, the versatile functional sites (e.g., open metal sites, functional groups in ligands and Lewis acidic/basic sites) can realize the specific recognition through different chemical reaction. Four, the tunability in optical properties of MOFs facilitates the development of fluorescent sensors with multiplex emissions. We do not discuss these topics here since many excellent reviews about the adsorption and separation of targets with porous MOFs are available elsewhere [54,55,56,57]. In this part, we mainly discussed the recently advancements of chemosensors for the direct detection of inorganic ions and small organic molecules.

### 3.1. Inorganic Ions

#### 3.1.1. Metal Ions

Inorganic metal ions, including heavy metal ions and transition metal ions, have an important role in environmental and ecological systems. Heavy metal ions (e.g., Cu^2+^, Fe^3+^, Hg^2+^ and Cr^3+^) can cause serious harm to the water and soil resources. Several transition metal ions are essential trace elements in biological metabolism. As a result, a large number of MOFs-based fluorescent sensors have been developed for the detection of metal ions in recent past years (Table 1). Due to the abundant functional groups of ligands in MOFs (e.g., carboxyl, amino, and phenolic hydroxyl group), metal ions with the unfilled *d* orbital can bind to the groups with the lone-pair of electrons to result in the quenched fluorescence of MOFs by PET. Several excellent reviews have been reported to summarize the progress of MOFs-based fluorescent assays for metal ions [58,59,60,61,62]. In this section, we mainly summarized the recently reported and representative literatures.

Copper is a fundamental metal element in biological systems due to its enzymatic property. Exposure to excessive concentrations of Cu^2+^ may cause neurodegenerative disorders such as Wilson’s disease and Parkinson’s disease. Several fluorescent MOFs-based Cu^2+^ sensors have been developed in a signal-off mode based on the excellent fluorescence quenching ability of Cu^2+^. For example, Chen et al. designed a ratiometric fluorescent probe for Cu^2+^ detection by integrating UiO-66(OH)_2_ with porphyrin-based MOF PCN-224 (Figure 2A) [63]. The green-emission UiO-66(OH)_2_ acted as the internal reference, while Cu^2+^ quenched the red emission of PCN-224 by binding to the porphyrin ligand with strong affinity. Cheng et al. applied fluorescent porphyrinic MOF-525 as the probe for signal-off detection of Cu^2+^ ions [64]. Because the sensitivity of “turn-off” sensors may be lower than that of “turn-on” sensors. Chen et al. reported “turn-on” fluorescent sensing of Cu^2+^ ions based on the porphyrinic MOFs-catalyzed Heck reaction [65]. As displayed in Figure 2B, porphyrin in PCN-222 was pre-inserted with Pb^2+^ ions. Cu^2+^ ions could replace Pb^2+^ ions due to the stronger binding affinity. Then, the released Pb^2+^ ions were in-situ reduced into PbNPs in MOFs and catalyzed the transformation of non-fluorescent aniline into fluorescent product via Heck cross-coupling reaction. More interestingly, ratiometric fluorescence MOFs-based sensors have been widely constructed because they can overcome the shortcomings of single-emission fluorescence methods. Xia et al. prepared dual-emission MOFs with dual ligands and used them for the ratiometric detection of Cu^2+^ ions in human serum [66]. As shown in Figure 2C, the ligand terephthalic acid (H_2_BDC) was used to sensitize Eu^3+^ ions via the antenna effect. 2,5-Diaminoterephthalic acid (H_2_DATA) with the emission at 465 nm could selectively identify the target. Cu^2+^ was coordinated with the -NH_2_ group of DATA, thus quenching the fluorescence of DATA via PET. For simultaneously monitoring of Cu^+^ and Cu^2+^ ions in biological samples, Zhang et al. reported a dual-channel fluorescence method for the speciation of both Cu^+^ and Cu^2+^ ions based on CDs-encapsulated Eu-MOFs (Figure 2D) [67]. In this work, fluorescent CDs were prepared from folic acid through a hydrothermal method and then in-situ encapsulated in Eu-MOFs. Cu^2+^ could quench the fluorescence of Eu-MOFs at λ_ex_/λ_em_ = 380/454 nm by replacing Eu^3+^ in the frameworks. Cu^+^ was interacted with bathocuproine disulfonate (BCS) and the product (BCS−Cu^+^) to quench the fluorescence of CDs at λ_ex_/λ_em_ = 275/615 nm via the IFE mechanism.

Aluminum is widely used in aluminum tableware, food packaging, and other production products. An excessive intake of aluminum in the body will disrupt the activities of the central nervous system, cause human dysfunction, and hinder the normal metabolism of calcium and phosphorus. Zhong et al. prepared two types of fluorescent curcumin@MOF-5 composites through post-adsorption and one-pot synthesis and applied them to detect Al^3+^ ions [68]. In this method, the fluorescence was quenched by the PET process, and Al^3+^ could inhibit the PET process between curcumin and MOFs-5, resulting in the restoration of curcumin fluorescence. Li et al. applied an AIE-active MOFs for sensing of Al^3+^ in a “turn-on” mode (Figure 3A) [69]. The fluorescence of Zn-MOF containing AIE ligands was quenched by the auxiliary ligand of 4,4’-bypyridine (Bpy) through the exciton migration. Al^3+^ was competitively coordinated with Bpy to recovery the fluorescence.

Fe^3+^ is an essential trace element in human nutrition. However, the abnormal level of Fe^3+^ may cause many health problems, such as anemia, diabetes, cardiovascular failure, liver laceration and so on. Due to the half-filled *d* orbits and the light absorption ability, MOFs (e.g., NTU-9 nanosheets, Tb-MOFs and dye@Zr-MOFs) were used to develop fluorescence Fe^3+^ sensor in which Fe^3+^ ions could quench the fluorescence of MOFs through the PET and/or IFE process [70,71,72,73]. For example, Qi et al. demonstrated that Fe^3+^ could quench the fluorescence of Tb-MOF via the IFE process and the inhibition of the energy transfer between linker and Tb^3+^ [74]. Recently, Yin et al. developed a fluorescent sensor for Fe^3+^ and HCHO detection based on the mixed-ligand MOFs [75]. As shown in Figure 3B, the prepared Zn-MOFs exhibited dual emission at 450 and 550 nm and the relative intensity was controlled by adjusting the ligand ratio and the excitation wavelength. Fe^3+^ could be coordinated with the amino group of the ligand 2-aminoterephthalic acid (BDC-NH_2_), thus quenching the fluorescence via the PET and IFE mechanism. HCHO could react with the amino group of BDC-NH_2_ to form a Shiff base, leading to the decrease of the fluorescence. Consequently, the yellow emission of the ligand of 2,5-dihydroxylterephthalic acid [BDC-(OH)_2_] at 550 nm increased. Additionally, Fe^3+^ ions can cause the collapse of the structure of Tb-MOFs and disturb the energy transfer from the ligand to Tb^3+^, resulting in the quenching of the fluorescence of Tb-MOFs [76].

As one of the most toxic heavy metals, mercury exists extensively in the ecosystem by the format of inorganic ion (HgX_2_, X = halide) or organic mercury [77]. Guo et al. prepared Eu^3+^-functionalized Ca-MOF through the post-synthesis modification and applied it for the detection of Hg^2+^ [78]. In this study, Hg^2+^ chelated with the soft Lewis base S in the ligand and inhibited the LMCT, leading to the decrease of the fluorescence signal of Eu^3+^ and the recovery of the emission of the ligand. Wang et al. reported the detection of Hg^2+^ and CH_3_Hg^+^ ions by using boric acid (BA)-functionalized Eu-MOF as the dual-signal probe (Figure 3C) [79]. The BA-Eu-MOF was prepared with 5-boronobezene-1,3-dicarboxylic acid (5-bop) as the ligand. The 5-bop ligands can sensitize the red emission of Eu^3+^ and provide the binding sites for Hg^2+^ and CH_3_Hg^+^. The electron-withdrawing effect of BA could passivate the antenna effect of 5-bop. Hg^2+^ and CH_3_Hg^+^ ions could replace the BA groups in the framework by transmetalation reaction, thus activating the antenna effect of 5-bop and resulting in the increase of red emission.

**Figure 3 biosensors-12-00928-f003:**
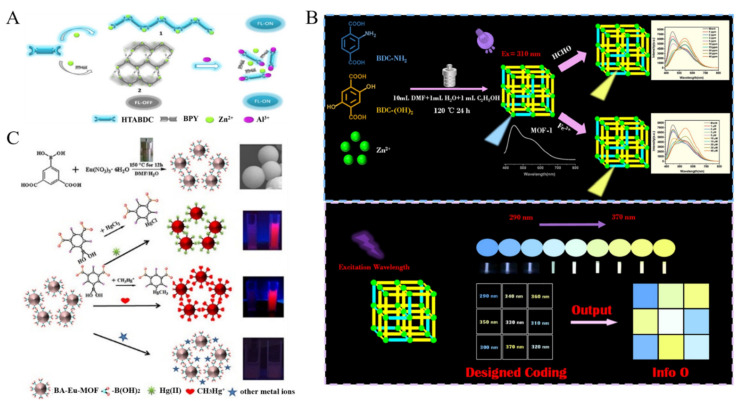
(**A**) Schematic illustration of the AIE-active MOFs-based sensor for the detection of Al^3+^. Reproduced with permission [69]. Copyright 2018, American Chemical Society. (**B**) Schematic illustration of BA-Eu-MOF synthesis and the sensing process of BA-Eu-MOF toward Hg^2+^ and CH_3_Hg^+^ ions based on transmetalation. Reproduced with permission [75]. Copyright 2016, American Chemical Society. (**C**) Schematic illustration of the design principle of dual-emission Eu-DATA/BDC MOFs and the ratiometric sensing and visible detection of Cu^2+^ ions in human serum. Reproduced with permission [79]. Copyright 2022, Elsevier.

Furthermore, Li et al. developed a Cd(II)-MOFs-based signal-on fluorescence sensor for Cr^3+^ detection [80]. Cr^3+^ could be chelated by the –COOH group on the ligand, improving the rigidity of the framework and resulting in the increased fluorescence. Liu’s prepared NH_3_ plasma-functionalized UiO-66-NH_2_ for the detection of U(VI) [81]. The functionalized UiO-66-NH_2_ showed high fluorescence intensity due to the large number of electron-donating groups. U(VI) could quench the fluorescence of MOFs through the static quenching effect by binding to the amino group.

pH plays an important regulatory role in physiological/pathologic processes. The abnormal intracellular pH is related to many diseases, such as cancer, stroke, and Alzheimer’s disease [82]. Several fluorescent MOFs have been reported to exhibit response toward different pH values, such as PCN-225(Zn), Y(III)-MOFs and Dy-MOF [83,84,85,86,87]. Typically, Chen et al. reported a dual-emission fluorescent MOF for monitoring the change of pH value with an excitation wavelength (Figure 4) [88]. In this work, the DBI-PEG-NH_2_-functionalized Fe_3_O_4_ NPs were first embedded to the MOF PCN-224. Then, rhodamine B isothiocyanate (RBITC) was conjugated onto the surface of Fe_3_O_4_ to form the RBITC-PCN nanoprobe. The covalent coupling of RBITC with DBI-PEG-NH_2_ prevented the leaching of RBITC. Under single excitation, the core-shell-like nanoprobe showed dual-emission from porphyrin of PCN and RBITC. H^+^ ions could enter into the pores rapidly to open the channels of MOFs. The RB-PCN nanoprobe could detect the pH change over a wide range. In the acidic pH (1.7 − 7.0), the nanoprobe showed two emissions from RBITC at 575 nm. However, it showed an emission at 641 nm for PCN-224 in the pH ranging from 7.0 to 11.3.

#### 3.1.2. Anions

Anions (e.g., Cl^−^, Br-, I^−^, SCN^−^ and NO_3_^−^) can replace the free counterions in MOFs via anion-exchange, thus changing the optical properties of MOFs. In addition, anions can restore the fluorescence of MOFs quenched by metal ions based on different interactions, such as high affinity interaction and chemical reaction, achieving a fluorescence “on-off-on” switch. A lot of MOFs-based fluorescence sensors have been fabricated to determine anions with satisfactory results (Table 1). Herein, we summarized only the recent advancement since several excellent reviews have been reported to discuss MOFs-based fluorescent anions sensors [89,90].

Fluoride ion (F^−^) is one of the most harmful elements in drinking water. Over-intake of F^−^ can result in dental fluorosis, kidney failure, or DNA damage. According to the Pearson hard−soft acid−base (HSAB) classification, F^−^ as a hard Lewis base can bind to Ln^3+^ ions as hard Lewis acids with high affinity. Thus, Ln-MOFs can be used to develop fluorescent sensors for F^−^ detection [91]. Zeng et al. synthesized a series of mixed Ln-MOFs with triazine-based planar 4,4,4”-s-triazine-2,4,6-triyltribenzoate (TATB) as the ligand and Tb/Eu ions as the centers (Figure 5A) [92]. The luminescent wavelength of Tb/Eu(TATB) MOFs could be regulated by adjusting the ratio of the two lanthanide metal ions. The host−guest interaction between F^−^ and MOF induced the signal change in color and luminescence intensity of MOFs, thus realizing the visual and ratiometric luminescent detection of F^−^. Che et al. reported a portable logic detector based on Eu-MOF for the detection of Eu^3+^ and F^−^ in groundwater [93]. In this study, Eu^3+^ combined with the amino groups on MOFs, leading to the aggregation of MOFs and the fluorescence quenching. F^−^ ions could replace Eu^3+^ to bind with the amino groups, resulting in the recovery of the fluorescence. Dalapati et al. prepared the pyrene-tagged UiO-66-NH_2_ through the post-synthetic modification and used it to detect F^−^ and H_2_PO_4_^−^ based on “turn-on” fluorescence mechanism [94]. In this work, the hydrogen bonding interactions between anions and pyrene units made the two π-stacked pyrene moieties closer and formed static pyrene excimers, resulting in the enhancement and blue shift of the fluorescence.

Hypochlorous acid (HClO) is a typical reactive oxygen species. Excessive endogenous HClO may cause oxidative stress, tissue damage and diseases. With dual ligands of BDC-NH_2_ and dipicolinic acid (DPA) and metal node Eu^3+^ ions as the precursors, a ratiometric fluorescent MOF probe was prepared (Figure 5B) [95]. The Eu-BDC-NH_2_/DPA MOF with exposed amino groups could be used to determine hypochlorous (ClO^−^) ions due to the formation of hydrogen bonds [95]. In this process, the blue fluorescence from BDC-NH_2_ was reduced, while the antenna effect emission from Eu^3+^ remained stable. Moreover, Tan et al. used MOFs as the scaffolds to develop a FRET system for the ratiometric detection of ClO^−^ ions [96]. In this work, CDs as the donor and curcumin as the acceptor were entrapped into MOFs for the construction of the FRET system. In the presence of ClO^−^, curcumin was oxidized and the FRET was blocked, resulting in the increase of the CDs fluorescence at 410 nm and the decrease of the curcumin (CCM) fluorescence at 585 nm.

Phosphate (PO_4_^3−^) plays an important role in many biological functions. An excessive content of PO_4_^3−^ in the human body is harmful to blood vessels and may accelerate aging [97]. PO_4_^3−^ has a significant quenching effect on the fluorescence of Ln-MOF or Ln-functionalized MOFs [98]. For example, Fan et al. used a Tb(III)-functionalized Zn-MOFs for fluorometric detection of PO_4_^3−^ [99]. In this work, the binding between PO_4_^3−^ and Tb^3+^ inhibited the antenna effect, leading to the decrease of Tb^3+^ emission. Recently, Shi et al. prepared dual-ligands Eu-MOF with dual emissions for the ratiometric/visual determination of PO_4_^3−^ [100]. As illustrated in Figure 5C, Eu-MOFs with benzene-1,3,5-tricarboxylate (BTC) and BDC-NH_2_ as ligands exhibited two emissions at 425 nm and 617 nm. PO_4_^3−^ could replace a part of the ligand to coordinate with Eu^3+^ and weakened the antenna effect, causing a reduced intensity at 617 nm and an enhanced signal at 425 nm.

Other anions can also be sensitively determined by fluorescent MOFs. For instance, Zhang et al. demonstrated that bromate could oxidize Fe^2+^ into Fe^3+^ in Eu-MOFs@Fe^2+^, thus quenching the fluorescence [101]. Karmakar et al. modified ZIF-90 with the dicyanovinyl (DCV) group for the specific recognition of CN^-^ (Figure 5D) [102]. The nucleophilic addition reaction between cyanide and DCV group resulted in the loss of conjugation and the decrease of fluorescence intensity. Jain et al. reported a boric-acid-functionalized Eu-MOFs-based sensor for Cr_2_O_7_^2-^ detection [103]. The red fluorescence of dual-emission MOFs was quenched and the blue emission was enhanced. Zhang et al. reported that Cr_2_O_7_^2-^ could quench the fluorescence of Zn-MOFs via the IFE mechanism [104].

**Figure 5 biosensors-12-00928-f005:**
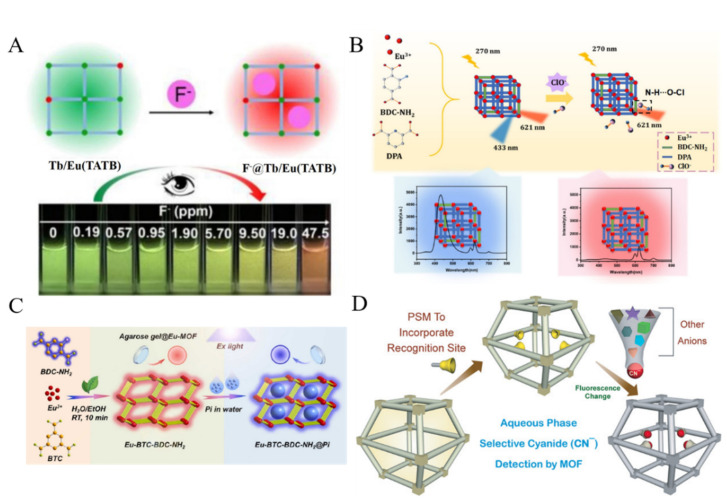
(**A**) Schematic illustration of visual detection of F^−^ using mixed Ln-MOFs with a smartphone. Reproduced with permission [92]. Copyright 2020, American Chemical Society. (**B**) Schematic illustration of the preparation of Eu-BDC-NH_2_/DPA and fluorescence spectra and photographs in the absence and presence of HClO. Reproduced with permission [95]. Copyright 2021, American Chemical Society. (**C**) Schematic illustration of the detection of PO_4_^3−^ based on Eu-BTC-BDC-NH_2_. Reproduced with permission [100]. Copyright 2022, Elsevier. (**D**) Schematic illustration of the post-synthetic modification in MOF leading to selective sensing of CN^−^. Reproduced with permission [102]. Copyright 2016, Wiley.

H_2_S is one of the reactive sulfur species that is generated from the decomposition of sulfur-containing organic molecules by microbes. Ma et al. prepared a Cu(II)-contained 3D porous nanoscale MOF (denoted as nano-MOF PAC) by using meso-tetrakis(4-carboxylphenyl)porphyrin (TCPP) as the ligand (Figure 6A) [105]. The reactive Cu(II) ions were introduced into the framework as the responding sites of H_2_S. The fluorescence of ligand in PAC was quenched by Cu(II) due to its paramagnetic characteristics. H_2_S could sequester Cu(II) from the porphyrin center to form CuS, thus turning on the fluorescence. With the nano-MOF as fluorescent probe, the signal-on in-situ detection of H_2_S in living cells was achieved. Gao et al. reported the detection of H_2_S with the ratiometric fluorescent MOF (rhodamine B (RhB)/UiO-66-N_3_) (Figure 6B) [106]. The nanoprobe exhibited dual emissions (425 nm and 575 nm) with a single excitation wavelength. The red fluorescence at 575 nm as the internal reference is attributed to RhB. The H_2_S-mediated biorthogonal reduction of azide to amine caused the increase in the blue fluorescence at 425 nm. The signal-on ratiometric sensing strategy could be used to image H_2_S in live cells.

**Table 1 biosensors-12-00928-t001:** Performances of MOFs-based fluorescent chemosensors for the detection of inorganic ions.

MOFs	Targets	Linear Range	LOD	Ref.
Cd-MOFs	Cr^3+^	0~3 mM	0.164 μM	[80]
UiO-66-NH_2_	U(VI)	0~1.2 μM	0.08 μM	[81]
UiO-66(OH)_2_@PCN-224	Cu^2+^	0~10 μM	0.068 nM	[63]
Porphyrinic Zr-MOF-525	Cu^2+^	1~250 nM	220 pM	[64]
Porphyrinic PCN-222-Pd(II)	Cu^2+^	0.05~2 μM	50 nM	[65]
Eu-MOF	Cu^2+^	1~40 μM	0.15 μM	[66]
CDs@Eu-MOF	Cu^+^ and Cu^2+^	0.5~20 μM and 0.5~20 μM	0.22 and 0.14 μM	[67]
CCM@MOF-5	Al^3+^	0~0.33 mM	2.84 μM	[68]
AIE-active Zn-MOFs	Al^3+^	Not reported	3.73 ppb	[69]
Tb-MOFs	Fe^3+^	0.33~33 μM	0.936 μM	[74]
Tb-MOFs	Fe^3+^	0~100 μM	0.35 μM	[71]
Zn-MOFs	Fe^3+^ and HCHO	1~40 μM and 1~40 ppm	0.58 μM and 370 ppb	[75]
RhB@Zr-MOF	Fe^3+^ and Cr_2_O_7_^2−^	0.01~1 mM and 1~100 μM	Not reported	[72]
Eu-modified Ga-MOFs	Hg^2+^	0.02~200 μM	2.6 nM	[78]
BA-Eu-MOF	Hg^2+^ and CH_3_Hg^+^	1~60 μM and 2~80 μM	220 and 440 nM	[79]
BODIPY@Eu-MOF	F^−^, H_2_O_2_ and glucose	0~30 μM, 0~6 μM and 0~6 μM	0.1737 μM, 6.22 nM and 6.92 nM	[91]
Tb^3+^ and Eu^3+^-MOFs	F^−^	0~1.9 ppm	96 ppb	[92]
Eu-MOFs	Eu^3+^ and F^−^	0~7.4 μM and 0~515 μM	0.2481 μM and 1.145 μM	[93]
Eu-MOFs	HClO	1~20 μM and 20~40 μM	37 nM	[95]
CDs/CCM@ZIF-8	HClO	0.1~50 μM	67 nM	[96]
Eu-MOFs	PO_4_^3−^	0.1~10 μM and 10~50 μM	0.07 μM	[100]
M-ZIF-90	CN^−^	0~0.1 mM	2 μM	[102]
UiO-66@COFs	PO_4_^3−^	0~30 μM	0.067 μM	[97]
Tb-modified Zn-MOFs	PO_4_^3−^	0.01~200 μM	4 nM	[99]
Eu-MOF@Fe^2+^	BrO_4_^−^	0~0.2 mM	3.7 μM	[101]
BA-Eu-MOFs	Cr_2_O_7_^2−^	0.1~3 μM	0.58 μM	[103]
Zn-MOFs	Cr_2_O_7_^2−^	0.3~20 μM	0.09 μM	[104]
RhB/UiO-66-N_3_	H_2_S	0.1~4 mM	82.4 μM	[106]

Abbreviation: CDs, carbon dots; CCM, curcumin; AIE, aggregation-induced emission; RhB, rhodamine B; BA, boric acid; BODIPY, boron-dipyrromethene; COFs, covalent organic frameworks.

### 3.2. Small Organic Molecules

Organic molecules with functional groups can coordinate with metal nodes or react with the linkers, finally changing the emission properties of MOFs. In addition, metal ions-mediated fluorescence “turn-off-on” switch can be utilized to detect small organic molecules. Based on these principles, many MOFs-based fluorescence sensors have been reported for the detection of a wide range of analytes, including nitro explosives, pesticides, antibiotics and other small molecules (Table 2) [107].

For homeland security, environmental and humanitarian implications, various fluorescent methods have been developed for explosive substances detection. Explosives with electron-deficient –NO_2_ groups are good electron acceptors that can interact with the electron-rich aromatic rings and functional groups with lone-pair electrons, leading to decreased fluorescence [108,109,110]. For this view, Zhang et al. prepared a 2D MOF (NENU-503) for the detection of nitroaromatic molecules with different numbers of –NO_2_ groups [111]. Wang et al. introduced a cationic dye into MOFs through ion-exchange process and used the fluorescent dye@bio-MOFs as the dual-emitting platforms for the detection of different kinds of nitro-explosive substances [112]. In this work, nitroaromatics could quench the fluorescence of MOFs, and aliphatic nitro-organics could increase the fluorescence via confinement-induced enhancement. Sun et al. in-situ encapsulated RhB molecules into Cd-MOFs and used the dual-emission of MOFs to detect 4-nitroaniline [113]. Sharma et al. synthesized a hydroxyl-functionalized indium MOF for the fluorescent detection of nitro-aromatic compounds [114]. The fluorescence was quenched due to the PET effect. With tetraphenylethylene, γ-cyclodextrin (γ-CD) and metal ion (K^+^) as the precursors, Qiu et al. synthesized γ-CD-MOF-K and used it to encapsulate the AIE molecules of tetraphenylethylene (Figure 7A) [115]. The resulting TPE@γ-CD-MOF-K composites exhibited a strong fluorescence emission. Attachment of nitro-aromatic compounds on the surface of TPE@γ-CD-MOF-K caused solid-state fluorescence quenching.

Chemical pesticides have been widely used around the world, including organophosphate pesticides and dinotefuran. When entering the surrounding environment, they may show an important influence on human health and life quality. MOFs have been used as fluorescent probes to detect various pesticides [116,117,118]. For example, Jiao et al. reported a fluorescent method for the detection of dinotefuran with Cd-MOFs, in which dinotefuran quenched the fluorescence of MOFs via the IFE and static quenching process [119]. Based on the host–guest interaction, Yu et al. reported the detection of pesticide glyphosate with tetra-pyridyl calix [4]arene decorated ultrathin 2D MOF nanosheet as the fluorescence probe (Figure 7B) [120]. Glyphosate could interact with the calix [4]arene (Calix) group on the surface of 3D layered MOF nanosheets, thus causing fluorescence enhancement. Dyes-encapsulated MOFs can also be used to detect pesticides. Zhang et al. used methylene blue (MB)-loaded Cd-MOF as the dual-emission probe for the detection of carbaryl that could enhance the fluorescence through the energy transfer and PET mechanisms [121]. Wei et al. prepared eosin Y (EY)-embedded Zr-MOFs for the ratiometric detection of nitenpyram (Figure 7C) [122]. In this study, the dual-emissive characteristics of EY@Zr-MOFs could be tuned by changing the loading quantity of EY. The FRET between EY@Zr-MOFs and nitenpyram blocked the energy transfer from pristine MOFs to EY, resulting in the decrease of the emission at 430 nm and 560 nm.

Aldehydes are one of the typical volatile organic compounds (VOCs), which can cause serious health problems. Wang et al. reported the detection of formaldehyde (FA) with N-propyl-4-hydrazine-naphthalimide (PHN)-embedded UiO-66-NH_2_ (PHN@MOF) as the ratiometric fluorescent probe (Figure 7D) [123]. The intrinsic fluorescence of UiO-66-NH_2_ offered a reference signal. FA was reacted with the hydrazine group of PHN through the surrounding confinement space provided by MOF.

**Figure 7 biosensors-12-00928-f007:**
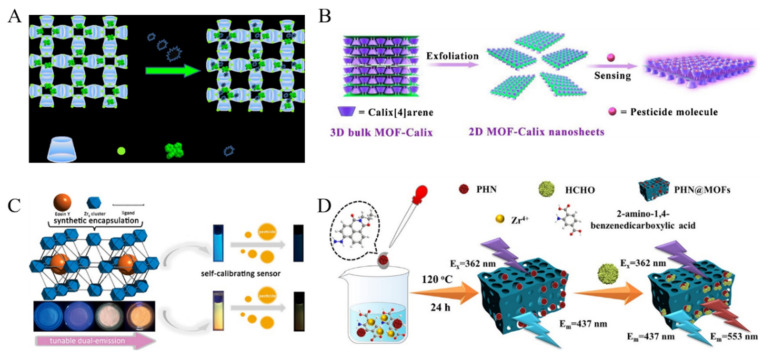
(**A**) Schematic illustration of the AIE-active MOFs for sensitive detection of explosives in liquid and solid phases. Reproduced with permission [115]. Copyright 2020, American Chemical Society. (**B**) Schematic illustration of the fabrication of 2D MOF-Calix nanosheets and sensitive detection of pesticide through host-guest chemistry. Reproduced with permission [120]. Copyright 2020, Elsevier. (**C**) Schematic illustration of EY-embedded Zr-MOFs as a dual-emitting built-in self-calibrating platform for nitenpyram detection. Reproduced with permission [122]. Copyright 2020, American Chemical Society. (**D**) Schematic illustration of the PHN@MOF probe for the detection of FA. Reproduced with permission [123]. Copyright 2021, American Chemical Society.

Antibiotics have been widely used in the treatment of infectious diseases by killing a variety of bacteria and parasites. The misuse of antibiotics seriously threaten the safety of water resources, food, and humans. Thus, many MOFs-based fluorescent sensing platforms have been constructed for the detection of antibiotics, including tetracycline antibiotics (TCs), nitrofuran and fluoroquinolone [124,125,126,127,128,129,130,131]. For instance, Chen et al. employed Zn-MOFs as fluorescent probes for the detection of TCs and NH^4+^ [132]. Qin et al. prepared two-dimensional ultrathin Cd-MOF/Tb^3+^ fluorescent nanosheets and applied them to detect cefxime antibiotic [133]. In this work, cefxime quenched the fluorescence of Cd-MOF/Tb^3+^ via the PET and IFE effects. To reduce environmental interferences, Gan et al. reported a ratiometric fluorescence sensor for visual detection of TC based on the nanohybrid of MOFs and lanthanide the coordination polymers [134]. As shown in Figure 8A, the coordination polymers assembled by Eu^3+^ and guanosine monophosphate (GMP) were in-situ synthesized on the surface of fluorescein-encapsulated MOF-5 (FSS@MOF-5). The resulting FSS@MOF-5/GMP-Eu composites exhibited two fluorescence (yellowish-green and red). In the presence of TC, the red emission from Eu^3+^ was enhanced due to the antenna effect between TC and Eu^3+^. The yellowish-green emission from FSS@MOF-5 kept constant as an internal reference. Under a UV lamp (365 nm), the fluorescence color of FSS@MOF-5/GMP-Eu changed from yellowish-green to red. Dual MOFs-based heterostructure composites (ZIF-8@PCN-128Y) were used to detect TC in milk and beef samples [135]. In this work, TC enriched by ZIF-8 quenched the fluorescence of PCN-128Y, attributing to the IFE and PET processes. To improve the selectivity of sensors toward TC antibiotics, Yu et al. found that the AIE effect of chlortetracycline (CTC) could be enhanced by binding with the zinc/pyromellitic acid-based MOF (Zn-BTEC) (Figure 8B) [136]. The CTC molecules were embedded into the porous Zn-BTEC to form assemblies or aggregates. The nanoprobe showed high sensitivity for CTC detection by discriminating it from other antibiotics with high specificity.

There are many toxic and cancerogenic substances in crops and food. For example, gossypol, a natural toxin existing in cottonseeds, shows a great risk to the safe consumption of cottonseed products. Rosi’s group found that Yb^3+^-based MOF (Yb-3,3″-diamino-1,1′:4′,1″-terphenyl-4,4″-dicarboxylic acid) could be used to detect gossypol with a low detection limit (Figure 8C) [137]. In this method, gossypol could sensitize Yb^3+^ and turn on its photoluminescence by facilitating the energy transfer and/or reacting with the amine-functionalized aromatic linker in MOF. The resulting Schiff base compounds caused the red shift of the wavelength. The proposed method could be further used for the design of novel probes for the detection of other aromatic molecules.

**Figure 8 biosensors-12-00928-f008:**
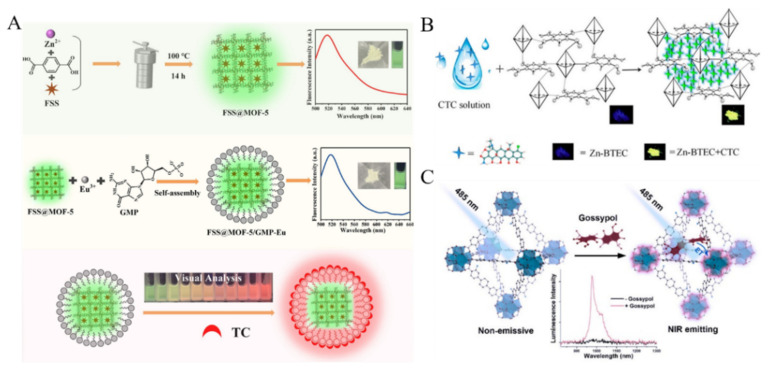
(**A**) Schematic illustration of the synthesis process of FSS@MOF-5/GMP-Eu for rapid and visual detection of TC. Reproduced with permission [134]. Copyright 2021, Elsevier. (**B**) Schematic illustration of sensing process for CTC using MOF Zn-BTEC as a fluorescent probe based on AIE mechanism. Reproduced with permission [136]. Copyright 2019, American Chemical Society. (**C**) Schematic illustration of fluorescent “turn-on” detection of gossypol using Ln^3+^-based MOFs and Ln^3+^ salts. Reproduced with permission [137]. Copyright 2020, American Chemical Society.

Biological molecules playing an important role in the pathophysiology can be recognized as the biomarkers for disease diagnosis. MOFs have been used to construct fluorescent sensors for the direct determination of biological molecules, such as hypoxanthine, uric acid, histidine and 3-nitrotyrosine [138,139,140,141,142,143]. For example, Zhang et al. used the mixed-crystal Ln-MOF as a fluorescent indicator for the detection of lysophosphatidic acid, a biomarker for ovarian cancer [144]. Wang et al. reported a UiO-66-NH_2_ MOF-based ratiometric fluorescent sensor for the detection of dopamine and reduced glutathione [145]. In this work, dopamine was reacted with polyethyleneimine to form copolymer, and it quenched the fluorescence of MOF via FRET. However, glutathione could inhibit the formation of copolymer and block the FRET, leading to the recovery of the fluorescence of MOFs. Reversibly, MOFs can also act as the acceptor to quench the fluorescent dye-labeled probes. Bhardwaj et al. reported a Tb-MOFs-based sensor for fluorescent detection of DPA, a biomarker for bacteria in feces [146]. In this work, the competitive interaction between Tb^3+^ ions and DPA inhibited the sensitization of Tb^3+^ ions in MOFs, leading to the decrease of the fluorescence. Li et al. reported a ratiometric fluorescent system for monitoring the main component of bacterial spore, DPA with CdS QDs-loaded ZIF-8 MOF as the sensing platform [147]. As shown in Figure 9A, Rhodamine 6G (Rho 6G) was encapsulated in the QDs/MOF nanocrystal as the internal reference. The fluorescence from QDs could be quenched by the capture of Eu^3+^ ions. Once the Eu^3+^ ions were sequestered by DPA, the fluorescence of QDs would be restored, while no signal change was observed for Rho 6G. The method showed a high sensitivity and excellent anti-interference ability for monitoring spore germination. Qu et al. reported the detection of creatinine by using 8-hydroxy-2-quinoline-carboxaldehyde (HQCA) and Al^3+^-modified MOFs (UiO-66-NH_2_) (Figure 9B) [148]. In this method, the formed creatinine-Al^3+^ Lewis acid-base complexes turned on the fluorescence by interrupting the energy transfer and electron transfer from UiO-HQCA to Al^3+^. With the rhodamine B (RhB)-encapsulated porous Zn(II)-MOF (DiCH_3_MOF-5) composite as the sensing platform, Guo et al. reported an “on-off-on” fluorescent switching method for the determination of Fe^3+^ and ascorbic acid (Figure 9C) [149]. The fluorescence of RhB@DiCH_3_MOF-5 could be quenched by Fe^3+^ due to the synergism of IFE and PET. However, addition of ascorbic acid to the quenched solution would recover the fluorescence due to the oxidation−reduction reaction. Benzene is one of the ubiquitous environmental pollutants. The trans,trans-muconic acid (tt-MA) is a suitable urinary biomarker for benzene at high levels. Qu and co-workers found that the bi-metal-loaded Eu(III)/Tb(III)@MOF-SO_3_^−^ showed a Tb(III)-induced luminescence of Eu(III) (Figure 9D) [150]. Based on the fluorescence quenching effect, the lanthanide-based MOF hybrids were used for the signal-off detection of tt-MA in urine.

**Table 2 biosensors-12-00928-t002:** Performances of MOFs-based fluorescent chemosensors for the detection of small organic molecules.

MOFs	Targets	Linear Range	LOD	Ref.
NH_2_–Cu-MOF	TNP	0.5~30 μM	80 nM	[108]
RhB@Cd-MOFs	4-nitroaniline	0~0.054 mM	43.06 μM	[113]
2D MOF-Calix	glyphosate	2.5~45 μM	2.25 μM	[120]
EY-Zr-MOF	nitenpyram	0~0.1 mM	0.94 μM	[122]
PHN@UiO-66-NH_2_	FA	1~3 and 3~4 μM	0.173 μM	[123]
UiO-67/Ce-PC	glyphosate	0.02~30 μg/mL	0.0062 μg/mL	[116]
Cd-MOFs	dinotefuran	0~130 μM	2.09 ppm	[119]
MB@Cd-MOF	carbaryl	0~90 μM	6.7 ng/mL	[121]
Zn-MOF	DOX, TET, OTC and CTC	0.001~46.67 μM for DOX, 0.001~53.33 μM for TET, OTC and CTC	0.56, 0.53, 0.58 and 0.86 nM	[124]
Zn-MOFs	ofloxacin	0~0.0215 mM	0.52 μM	[125]
Cd-MOFs	NFT and NFZ	4~18 nM	0.15 and 0.29 nM	[126]
Eu-MOFs	BRH and TET	0.5~320 μM and 0.05~160 μM	78 nM and 17 nM	[128]
Zn-MOFs	TEA, TET and NB	5~35 μM, 1~7.5 μM and 2~15 μM	1.07 μM, 0.1 μM and 0.2 μM	[129]
Zn-MOFs	OTC	0.02~13 μM	0.017 μM	[132]
Tb^3+^-modified Cd-MOFs	CFX	0~6 μM	26.7 nM	[133]
FSS@MOF-5/GMP-Eu	TET	0~20 μM	18.5 nM	[134]
ZIF-8@PCN-128Y	TET	0.4~200 μM	60 nM	[135]
Zn-BTEC MOFs	CTC	0~8 μM	28 nM	[136]
Yb-NH_2_-TPDC MOFs	gossypol	25~100 μg/mL	25 μg/mL	[137]
NH_2_-Cu-MOF	hypoxanthine	10~2000 μM	3.93 μM	[138]
Pyrene-modified Hf-UiO-66	UA	0~30 μM	1.4 μM	[139]
Eu^3+^-modified Mn-MOFs	histidine	0~30 μM	0.23 μM	[140]
Eu/Bi-MOF	histidine	0.001~10 mM	0.18 μM	[141]
Zn-MOFs	3-nitrotyrosine	0~4 μM	0.3099 μM	[142]
GNR and QD-embeded MOFs	BA	0.002~5 ppm	1.2 ppb	[143]
Tb-MOFs	DPA	0.001~5 μM	0.04 nM	[146]
CdS QDs@ZIF-8	DPA	0.1~150 μM	67 nM	[147]
HQCA-modified UiO-66-NH_2_	creatinine	0.05~200 μM	4.7 nM	[148]
RhB@Di-MOF	Fe^3+^ and AA	1~10 μM and 1~25 μM	0.36 μM and 0.31 μM	[149]
Eu(III)/Tb(III)@MOF-SO_3_^−^	*tt*-MA	0~20 μg/mL	0.1 μg/mL	[150]

Abbreviation: TNP, 2,4,6-trinitrophenol; RhB, rhodamine B; Calix, calix [4]arene; EY, eosin Y; PHN, N-propyl-4-hydrazinenaphthalimide; FA, formaldehyde; Ce-PC, porous carbon materials derived from Ce-MOF; MB, methylene blue; DOX, doxycycline; TET, tetracycline; OTC, oxytetracycline; CTC, chlortetracycline; NFT, nitrofurantoin; NFZ, nitrofurancillin; BRH, berberine hydrochloride; TEA, triethylamine; CFX, cefxime; FSS, fluorescein; GMP, guanosine monophosphate; BTEC, pyromellitic acid; NH_2_-TPDC, 3,3″-diamino-1,1′:4′,1″-terphenyl-4,4″-dicarboxylic acid; UA, uric acid; BA, benzaldehyde; GNR, gold nanorod; QD, quantum dot; DPA, dipicolinic acid; HQCA, 8-hydroxy-2-quinolinecarboxaldehyde; tt-MA, trans,trans-muconic acid.

For some solid pharmaceuticals and chemical products, water may be a contaminant and impurity. Trace water may affect the stability and efficacy of drugs. Several MOFs-based fluorescent sensors have been developed for water determination. For instance, Yin et al. reported the ratiometric fluorescence detection of water in organic solvents based on Ru(bpy)_3_^2+^-encapsulated MIL-101(Al)−NH_2_ [151]. Water facilitated the protonation of the ligand BDC-NH_2_, resulting in the shift of the fluorescence of MOFs. Zhou et al. presented a ratiometric and turn-on fluorescence method for water detection by using Eu-MOFs as the probes [152]. Water could switch on the intramolecular charge transfer from the electron donating amino group to the electron-withdrawing carboxyl group, leading to the enhancement of the fluorescence from ligand. However, these sensors are limited to the determination of water in organic solvents. To realize the analysis of water in solid samples, Yu et al. designed a portable analytical device based on Eu-dipicolinic acid/2-aminophthalic acid (Eu-DPA/PTA-NH_2_) [153] MOFs. As shown in Figure 10, a one-to-two logic gate was developed for determining water content and the two fluorescence signals as the input and output signals. MOFs were deposited on the fiber paper to prepare a microsensor. When the water content increased, the fluorescence of PTA-NH_2_ increased and the fluorescence of Eu^3+^ declined. Under the illustration of 254 nm UV light, the color of paper changed from red to blue. Furthermore, Chen et al. developed a ratiometric fluorescent and colorimetric method for the detection of methanol using bimetallic Ln-MOFs [154]. In this work, methanol could improve the LMCT efficiency from the ligand to Tb^3+^, resulting in the increase of the fluorescence of Tb^3+^ ions.

## 4. Biosensors

Chemicals and metabolites can provide important information about human healthy. Thus, it is meaningful to monitor their contents from the biological samples. Biomarkers can reflect the relationship between biological systems and external chemical, physical and biological factors. Thus, considerable effort has been devoted to develop fluorescent MOFs-based biosensors for the determination of biomarkers (Table 3) [155,156,157,158,159]. Among them, a variety of DNA-based switch fluorescent biosensors have been extensively developed for the detection of various biomarkers because of the intrinsic specificity and predictability, including small biomolecules, nucleic acids, enzymes, proteins and tumor cells. In common switch fluorescent DNA/MOFs-based biosensors, dye-labeled DNA/aptamers are adsorbed on the surface of MOFs through the π–π stacking, electrostatic, coordination, and hydrogen bonding interactions. Then, the fluorescence was efficiently quenched by MOFs via PET or FRET process in a “turn-off” mode [160]. In the presence of a target, the higher affinity between target and DNA/aptamer may result in the release of DNA/aptamer from the surface of MOFs, thus recovering the fluorescence. In this part, we highlighted the MOFs-based for the detection of biomarkers including small biomolecules, nucleic acids, enzymes, proteins and others.

### 4.1. Small Biomolecules

Nucleotides as the energy supply for human body have a critical influence on the regulation of various cellular metabolic processes. Extensive efforts have been focused on the fabrication of MOFs-based fluorescent biosensors for the detection of nucleotides [161,162,163]. For example, Qu et al. developed a fluorometric aptasensor for adenosine triphosphate (ATP) detection [164]. In this method, AuNPs were used to quench the emission of Tb-MOF by FRET. Under the condition of high concentration of salts, AuNPs would assembly into aggregates, thus losing the fluorescence quenching ability. In the presence of ATP, the aptamer specifically bound to ATP, and the complex was adsorbed onto the AuNPs to prevent the formation of aggregates, resulting in the decrease of the fluorescence. However, due to the similar structure and binding mode of ATP with its analogues, it is difficult to selectively respond to certain nucleotides with this method. To further improve the discrimination ability, Wang et al. synthesized a series of bimetallic Co_x_Zn_100-x_-ZIF (x = 0~100) MOFs and investigated their interactions with different nucleoside triphosphates (Figure 11A) [165]. It was found that Co_50_Zn_50_-ZIF and Co_80_Zn_20_-ZIF could recognize guanosine triphosphate (GTP) and ATP with high specificity. However, Co_65_Zn_35_-ZIF and Co_20_Zn_80_-ZIF could bind with both ATP and GTP. The resulting MOFs could quench the fluorescence of dye-labeled DNA. The target nucleoside triphosphate was then determined by displacing the attached DNA probe from the MOF surface to recover the fluorescence.

Due to its simplicity and specificity, aptamers against different molecules have been designed and used to develop fluorescent biosensors by combining with MOFs [166]. For example, Lu et al. reported a bimetallic MOFs-based fluorescent aptasensor for the detection of chloramphenicol [167]. Amalraj et al. developed a dual-mode MOFs-based fluorescent aptasensor for simultaneous determination of 17β-estradiol and chloramphenicol using two different dyes-labeled aptamers [168]. Ochratoxin A (OTA) is primarily generated through the consumption of improperly stored food products and uptake of OTA may cause organ damage. Recently, Li et al. developed a Zr-MOFs-based fluorescent and electrochemical dual-channel biosensor for OTA detection [169]. As displayed in Figure 11B, in the presence of OTA, its aptamer was released from the surface of Zr-MOFs due to the specific interaction between them. Thus, the fluorescence of dye-labeled aptamer was enhanced and the decreased amount of aptamer on the electrode surface resulted in a reduced electrochemical signal.

**Figure 11 biosensors-12-00928-f011:**
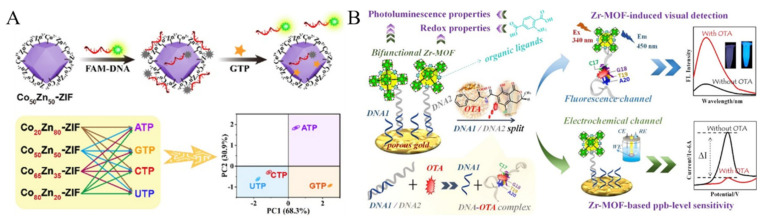
(**A**) Schematic illustration of DNA coated CoZn-ZIF MOFs for fluorescent sensing GTP. Reproduced with permission [165]. Copyright 2022, Elsevier. (**B**) Schematic illustration of the dual-channel Zr-MOFs-based detection strategy of OTA. Reproduced with permission [169]. Copyright 2022, American Chemical Society.

### 4.2. Nucleic Acids

Based on the canonical Watson–Crick model, DNA is a type of natural probe for selective detection of nucleic acids. MOFs can selectively adsorb ssDNA against dsDNA due to the difference in the adsorption affinity, thus quenching the fluorescence of dye-labeled ssDNA via PET or FRET [170,171]. For example, Chen et al. for the first time reported a Cu-MOF-based biosensor for the fluorescent detection of duplex DNA (HIV 16-bp oligopyrimidine oligopurine proviral DNA) [172]. In this study, the duplex DNA target could preferentially bind to the dye-labeled probe triplex-forming oligonucleotide (TFO), leading to the release of TFO from MOFs and the restoration of the fluorescence. Zhao et al. prepared ultrathin 2D Zn-TCPP MOF nanosheets through a surfactant-assisted synthetic method and used them to construct a fluorescent biosensor for the detection of DNA [173]. In this work, Zn-TCPP MOF with a π-electron conjugated system could adsorb the dye-labeled ssDNA and quench the fluorescence via FRET. The influenza A virus subtype H5N1 gene hybridized with ssDNA to form dsDNA, thus recovering the fluorescence. Recently, Zhang et al. investigated the interaction between ultrathin 2D Zr-BTB MOF nanosheets and fluorophore-labeled dsDNA as well as ssDNA by molecular dynamics simulations (Figure 12A) [174]. The nanosheets composed of Zr-O clusters and 1,3,5-benzenetribenzoate were prepared by a bottom-up method. Based on the difference in the affinity of dsDNA and ssDNA toward the Zr-BTB MOF nanosheets, the target DNA could be determined with high sensitivity and specificity. Besides the direct adsorption, dye-labeled DNA was covalently immobilized on MOFs, leading to the quenching of the fluorescence. Han et al. developed a fluorescent biosensor for the ratiometric detection of miRNAs based on UiO-66-NH_2_ and triple helix molecular switch (THMS) [175]. As shown in Figure 12B, the pre-formed THMS was modified on the amino-functionalized MOFs. The emission from UiO-66-NH_2_ at 440 nm was selected as the internal reference and that from FAM at 525 nm was related to the detection of target miRNA-203. The target miRNA-203 could hybridize with the dye-labeled probe, causing it to move away from MOFs to restore the quenched fluorescence.

By using different dye-labeled DNA probes, multiplex detection of DNA can be realized by monitoring the corresponding emission at different wavelengths [176]. For instance, Ye et al. developed a Cu-MOFs-based fluorescent biosensor for multiplexed detection of DNA with two carboxyfluorescein (FAM) and 5(6)-carboxyrhodamine triethylammonium salt (ROX)-modified probes [177]. Yang et al. synthesized a 3D zwitterionic Cu-MOFs for the selective fluorescent detection of human immunodeficiency virus 1 (HIV 1) ds-DNA and Sudan virus RNA [178].

MOFs with intrinsic fluorescence can be used as the electron or energy donor to develop biosensor for DNA detection. Afzalinia et al. developed a fluorescent miRNA biosensor by using Ln-MOFs as the donors and silver nanoparticles (AgNPs) as the acceptors [179]. As shown in Figure 12C, fluorescent Ln-MOFs and AgNPs were modified with two DNA probes that were partially complementary to target miRNA-155. In the presence of miRNA-155, the energy donor−acceptor probes were linked and the fluorescence of Ln-MOFs were quenched through FRET. G-Quadruplexes (G4s), a typical noncanonical conformation of guanine-rich DNA and RNA fragments, can regulate the transcriptional functions of the genome. Kouzegaran et al. reported the detection of human telomeric G4s DNA based on a hemin-modified fluorescent Ce-MOFs [180]. In this work, the hemin molecules were adsorbed on the pores and surface of Ce-MOF to quench the fluorescence. The target H-Telo G4s bound to hemin with the high affinity, resulting in the recovery of the fluorescence.

Due to their excellent properties of large specific surface area with well-defined porosity, MOFs can be employed as nanocarriers to load a large number of dyes and release the guest molecules under certain stimulus (e.g., targets, hypoxia and acidic pH). For this view, Wu et al. reported a fluorescent biosensor for the multicolor detection of DNA by using MOFs to load different dyes (Figure 12D) [181]. In this study, different DNA hairpins were used to cap the pores, preventing the leakage of dyes due to the steric-hindrance effect. In the presence of targets, the DNA hairpins hybridized with its complementary DNA targets to release the fluorophores from the MOFs’ pores, producing an enhanced fluorescence signal. However, the single signal mode may suffer from the interference of other substances in complex samples. To improve sensitivity and selectivity, Han et al. reported a DNA/MOF-based ratiometric fluorescent system for miRNA detection by FRET [182]. In this work, UiO-66-NH_2_ MOFs were used as the carriers to encapsulate RhB and capture DNA via the formation of Zr-O-P bonds. The DNA-modified MOFs could further encapsulate Thiazole Orange (TO) to form DNA-RhB@UiO-66-NH_2_. The target miRNA could bind with the ratiometric fluorescent DNA-RhB@UiO-66-NH_2_ probe by strand displacement reaction, thus restoring the fluorescence of TO.

**Figure 12 biosensors-12-00928-f012:**
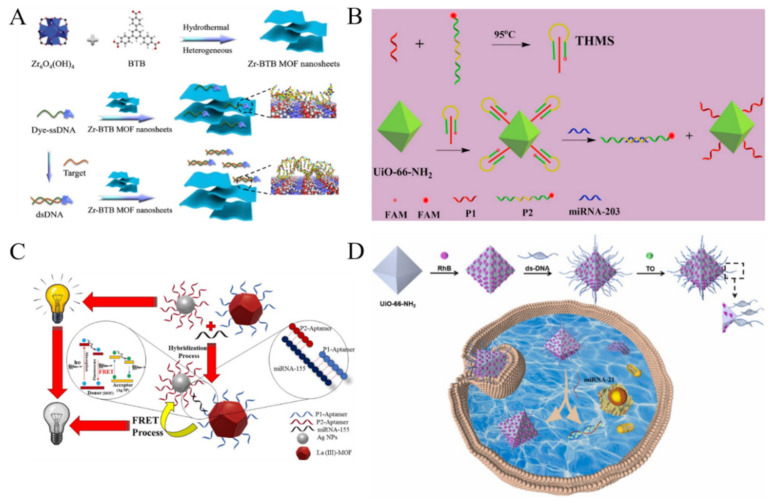
(**A**) Schematic illustration of preparation of Zr-BTB MOF nanosheets and projected interaction between nanosheets with ssDNA and dsDNA. Reproduced with permission [174]. Copyright 2020, American Chemical Society. (**B**) Schematic illustration of UiO-66-integrated programmable DNA triple-helix molecular switch for tumor-related miRNA-203 ratiometric detection. Reproduced with permission [175]. Copyright 2021, Elsevier. (**C**) Schematic illustration of fluorescence quenching-based detection of miRNA-155 by the FRET process. Reproduced with permission [179]. Copyright 2020, American Chemical Society. (**D**) Schematic illustration of fabrication of the probe (DNA-RhB@UiO-66-NH_2_ + TO) and reaction mechanism. Reproduced with permission [182]. Copyright 2022, Elsevier.

Dysregulated expression of miRNAs is associated with different diseases including cancers. Real-time imaging of miRNAs with low expression level in vivo would provide valuable information for disease diagnosis and evaluation of drug efficiency. MOFs can be used as the carriers to fabricate the stimuli-responsive cargo delivery nanosystems and efficiently deliver nucleases and detection probes to cytoplasm [183]. For example, Zhang et al. reported an enzymatic amplification strategy for the detection of miRNA in living cells by using pH-sensitive MOF (ZIF-8) nanoparticles to deliver ϕ29 DNA polymerase and detection probes (Figure 13A) [184]. After entering cells, the ZIF-8 nanoparticles could be dissolved due to the low pH environment, which led to the release of ϕ29DP and DNA probes. Then, the target miRNA triggered a rolling circle amplification (RCA) with the aid of ϕ29DP. The autonomously produced Mg^2+^-dependent DNAzymes could cleave the fluorogenic substrates, thus generating a readout fluorescence signal. Meng et al. reported the in vivo detection of aberrant miRNA by using a hypoxia-responsive Cu-MOF to load the signal strand block Cu-specific DNAzyme precursors and substrate strands (Figure 13B) [185]. The tumor microenvironment promoted the dissolution of DNA@Cu-MOF, leading to the release of extensive Cu^2+^ ions, DNAzyme precursors, and substrate strands. The aberrant target miRNA could displace the signal strand by toehold-mediated strand displacement hybridization, thus recovering the fluorescence signal of Cy3-labeled block DNA. More importantly, this reaction would trigger the Cu-specific DNAzyme signal amplification and cause the release of an increasing number of Cy3-labeled DNA stands. Thus, the work proposed a novel method for aberrant miRNA-related hypoxic tumor diagnosis.

### 4.3. Enzymes

Enzymes play an important role in almost all organs by participating in different biological processes. Overexpressed enzymes are closely related to various diseases. Therefore, the development of effective methods for enzyme activities is of great importance for the early diagnosis of diseases. The fluorescence of MOFs can be modulated by enzyme catalysis in which the substrate or the product can quench the fluorescence of MOFs via different mechanisms. For example, Yu et al. used the dual-emission Ln-MOFs for the detection of alkaline phosphatase (ALP) and demonstrated that ALP-based enzymatic products of phosphate ions (PO_4_^3-^) could quench the emission of Tb^3+^ ions and enhance the fluorescence of ligands by blocking the energy transfer from ligand to Tb^3+^ [186]. Guo et al. found that the RhB-encapsulated porous crystalline Zn(II)-MOF (RhB@MOF-5) could be used to detect β-Glucuronidase (β-GCU) (Figure 14A) [187]. In this work, the RhB@MOF-5 nanoprobe showed a yellow-green emission of RhB at 552 nm. Based on the synergistic effect of IFE and static quenching effect (SQE), the fluorescence of RhB could be quenched by 4-nitrophenyl-β-D-glucuronide (PNPG), the substrate of β-GCU. The enzymatic hydrolysis of PNPG into p-nitrophenol caused the recovery of fluorescence, thus achieving the signal-on detection of β-GCU in human serum.

Taking advantage of the quenching ability of metal ions, enzyme activities can be determined through the capture of metal ions from MOFs by their products or substrates. For example, Chen et al. reported the detection of glutathione S-transferase (GST) with brightly red emissive ZIF-8@QDs (Figure 14B) [188]. The nanoprobes exhibited the advantages of MOFs, and the assembly of Hg-ZnSe QDs and ZIF-8 MOFs enhanced the fluorescence of QDs. Cu^2+^ ions were attached onto the surface of ZIF-8@QDs, thus leading to the fluorescence quenching. The addition of glutathione (GSH) recovered the fluorescence of ZIF-8@QDs by the interaction of Cu^2+^ and GSH. However, GST could catalyze the reaction between GSH and 1-chloro-2,4-dinitrobenzene (CDNB), thus preventing the release of Cu^2+^ ions from the ZIF-8@QDs.

Enzymatic products can further react with other reagents to in-situ form complexes that can change the fluorescence of MOFs. Yu et al. reported a dual-response biosensor, the detection of tyrosinase (TYR) monophenolase activity by integration of fluorescent polymer dots and luminescent lanthanide MOF (Ln-MOF) (Figure 14C) [189]. In this study, monoaromatic ligand DPA was used as the linker for the preparation of Eu^3+^-based Ln-MOF. In alkaline boric acid buffer, L-tyrosine was converted into boric acid-levodopa by TYR monophenolase. With the aid of Eu-DPA, BA-levodopa was initiated by diethylaminepropyltrimethoxysilane (DAMO), thus promoting the formation of BA-levodopa polymer dots. This reaction could turn on a strong blue fluorescence and quench the red fluorescence of Eu-DPA via enhanced PET.

Moreover, enzyme products can regulate the surface, chemical and structure properties of MOFs by tuning the formation of chemical bonds [190,191]. Li et al. reported a luminescent MOFs-based label-free assay of polyphenol oxidase (PPO) [192]. In this study, the fluorescence of La-based MOFs was quenched by the enzymatic product of *o*-benzoquinone through the Michael-type addition reaction between o-quinone and the free amino group of MOFs. Wang et al. proposed a strategy for monitoring apyrase activity by mediating the catalytic and fluorescence quenching abilities of nanoscale MOFs through coordination-driven self-assembly (Figure 14D) [193]. Platinum nanoparticles (PtNPs) were loaded onto the commonly used MIL-88B-NH_2_ Fe-MOF. The catalytic activity of Pt/MIL-88B-NH_2_ nanozyme was enhanced due to the binding of ATP. Meanwhile, the fluorophore-labeled DNA could adsorb onto the surface of Pt/MIL-88B-NH_2,_ thus quenching the fluorescence signal. Enzymatic products, such as phosphates, small DNA strands and H_2_O_2_, could lead to the release of ATP and fluorophore-labeled DNA, thus allowing for the detection of apyrase and alkaline phosphatase. Cai et al. encapsulated gold nanoclusters (AuNCs) into MOFs and used the fluorescent MOFs to detect organophosphorus pesticides with the aid of acetylcholinesterase (AChE) and choline oxidase (CHO) [194]. In this study, AuNCs were confined in MOFs to show an enhanced fluorescence due to the AIE effect. Under the sequential catalysis of AChE and CHO in the presence of acetylcholine, the produced H_2_O_2_ could destroy the structure of MOFs and weaken the restraint on AuNCs, resulting in decreased fluorescence. However, pesticides could inhibit the enzyme activity, thus protecting MOFs from the decomposition and leading to the unchanged fluorescence.

**Figure 14 biosensors-12-00928-f014:**
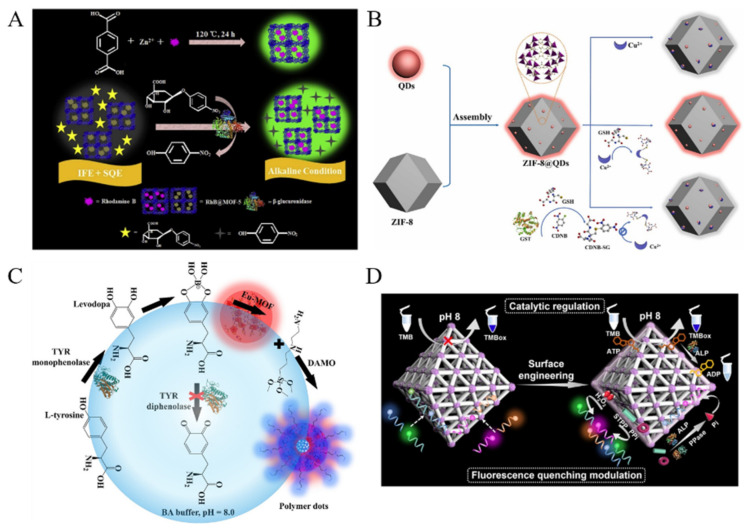
(**A**) Schematic illustration of the preparation procedure of RhB@MOF-5 by one-pot synthesis approach and the mechanism for β-GCU detection based on the synergistic effect of IFE and SQE. Reproduced with permission [187]. Copyright 2019, Elsevier. (**B**) Schematic illustration of the preparation of ZIF-8@QDs nanoprobe and its detection principle for GST activity. Reproduced with permission [188]. Copyright 2022, Elsevier. (**C**) Schematic illustration of fluorescence on/off dual-response sensing of TYR monophenolase activity based on Eu-MOF-assisted boric acid-levodopa polymer dots. Reproduced with permission [189]. Copyright 2022, Elsevier. (**D**) Schematic illustration of enzyme-modulated catalytic and fluorescence quenching of nanoscale MOFs. Reproduced with permission [193]. Copyright 2022, American Chemical Society.

### 4.4. Proteins

Sensitive and specific determination of proteins is of great importance in early diagnosis, prognosis and therapeutic evaluation. MOFs-based fluorescent aptasensors have also been developed for protein detection. For example, Zhang et al. reported a fluorescence proximity assay for the detection of prostate specific antigen (PSA) by using MOFs as the platforms [195]. Huang et al. reported a no-wash fluorescence biosensor for ratiometric detection of PSA with the nanohybrid of MOF@AuNP@graphene oxide as the nanoquencher [196]. Zhang et al. reported the fluorescent detection and imaging of phosphorylation and glycosylation based on the Zr(IV)-phosphate and boronic acid-diol interactions, respectively (Figure 15) [197]. Phosphate could interact with the metal node Zr(IV) of UIO-66-NH_2_ to interrupt the metal-ligand charge transfer, restoring the fluorescence. Boronic acid anchored on the framework could react with alizarin red by the formation of borate ester bonds. The embedded alizarin red showed poor fluorescence. The release of alizarin red from the framework by the competitive reaction with glycosyl turned on the fluorescence. In addition, Wei et al. developed a MOFs-based fluorescent biosensor for H_5_N_1_ antibody detection [198]. In the absence of a target, exonuclease I (Exo I) catalyzed the hydrolysis of the DNA probe at the 3’-terminus and the fluorophore was released from the MOF, resulting in the recovery of the fluorescence. H_5_N_1_ antibody could bind with the antigen conjugating at the end of the DNA, and it protected the DNA from the hydrolysis, leading to the adsorption of DNA on MOFs and the quenching of the fluorescence.

### 4.5. Others

Exosomes derived from tumor cells carry important molecular information from parent tumor cells. Thus, increasing levels of exosomes are closely related to the occurrence and progress of tumors. Exosomes could be detected by using magnetic bead-modified MOFs, such as UiO-66 to extract the target [199]. As shown in Figure 16A, the fluorescent probe of UiO-66-NH_2_ modified with anti-epithelial cell adhesion molecule (anti-EpCAM) was used as the signal reporter to recognize the EpCAM-positive exosome. Circulating tumor cells are shed from metastatic sites or primary tumors to peripheral blood and travel in the blood vessels, which is regarded as the main reason of metastatic spread. Circulating tumor cells from the blood of cancer patients could be determined with rhodamine 6G (Rho 6G)-entrapped MOF probe [200]. The probe denoted as MOF-Rho 6G-DNA was prepared by connecting sulfhydrylated DNA (Arm-DNA), entrapping Rho 6G and modifying EpCAM hairpin DNA onto the MOF. The binding of CTCs with hairpin DNA opened the hairpin structure and thus induced its conformational change, causing the release of entrapped Rho 6G molecules. Moreover, BA-functionalized MOF named Zr-UiO-66-B(OH)_2_ has been used for fluorescence detection of bacteria (Figure 16B) [201]. Zr-UiO-66-B(OH)_2_ was attached onto the bacteria surface by the interaction of BA and glycolipid outside the bacteria, in the presence of which the fluorescence of Zr-UiO-66-B(OH)_2_ was intensified by converting -B(OH)_2_ group into -OH group.

**Table 3 biosensors-12-00928-t003:** Performances of different MOFs-based fluorescent biosensors.

Probes	Targets	Linear Range	LOD	Ref.
Tb-MOFs and aptamer-modified AuNPs	ATP	0.5~10 μM	0.32 μM	[161]
ZIF-67 and FAM-aptamer	ATP	0.03~30 μM	29 nM	[162]
Cr-MIL-101 and FAM-aptamer	ATP	5~400 μM	1.7 μM	[163]
Tb-MOFs, aptamer and AuNPs	ATP	0.05~10 μM	23 nM	[164]
Co_x_Zn_100-x_-ZIF (x = 0–100) and FAM-aptamer	GTP	0~50 μM	0.13 μM	[165]
UiO-66-NH_2_ and TAMRA-aptamer	AFB1	0~180 ng/mL	0.35 ng/mL	[166]
Cu/UiO-66 and ROX-aptamer	CAP	0.2~10 nM	0.09 nM	[167]
MOF−MoS_2_ and ROX/TAMRA-aptamer	CAP and 17E	0~5 nM	200 and 180 pM	[168]
Zr-MOFs and aptamer	OTA	0.10~160 pg/mL	0.051 pg/mL	[169]
Cu-MOFs and FAM-aptamer	H5N1 antibody	0.005~1 μM	1.6 nM	[198]
Cu-MOFs and FAM-aptamer	dsDNA	4~200 nM	1.3 nM	[172]
UiO-66-NH_2_ and FAM-DNA	miRNA	1~160 nM	400 pM	[175]
DNA-modified Ln-MOFs and DNA-modified AgNPs	miRNA-155	0.0027~0.01 pM	5.2 fM	[179]
Tb-MOFs	ALP	0~8 mU/mL	0.002 mU/mL	[186]
RhB@MOF-5	β-glucuronidase	0.1~10 U/L	0.03 U/L	[187]
Cu@Eu-BTC MOFs	ALP	0.3~24 mU/mL	0.02 mU/mL	[190]
Ln-MOFs	PPO	0.001~0.1 mU/mL	0.12 mU/mL	[192]
EpCAM-modified UiO-66-NH_2_	exosomes	168~1 × 10^6^ particles/μL	16.72 particles/μL	[199]
Zr-UiO-66-B(OH)_2_	*E. coli*	5~2.5 × 10^4^ CFU/mL	1 CFU/mL	[201]

Abbreviation: ATP, adenosine triphosphate; AuNPs, gold nanoparticles; FAM, 5-carboxyfluorescein; GTP, guanosine triphosphate; TAMRA, tetramethylrhodamine; AFB1, aflatoxin B1; ROX, 6-carboxy-x-rhodamine; CAP, chloramphenicol; 17E, 17β-estradiol; OTA, ochratoxin A; ALP, alkaline phosphatase; RhB, rhodamine B; QD, quantum dot; BTC, 1,3,5-benzenetricarboxylic acid; PPO, polyphenol oxidase; *E. coli*, *Escherichia coli*.

## 5. Conclusions

In summary, we presented a brief overview of the current status of MOFs-based fluorescent sensing platforms via diverse sensing mechanisms. MOFs exhibiting intrinsic emission or encapsulated with fluorescent species can be used as fluorescent probes for the detection of various targets. The excellent results of MOFs-based sensing platforms can be attributed to their distinguished characters of large surface area, high porosity and tunable optical properties. However, MOF-based fluorescent chemosensors and biosensors are still limited by several bottlenecks. For example, some MOFs involved in this review show low stability against moisture/aqueous conditions and prone to breakdown under certain harsh conditions. Modifying the linker with water-repelling (hydrophobic) functional groups may be an effective way to improve stability. Second, the poor solubility and inadequate dispersiveness may decrease sensitivity and limit the applications of MOFs in portable sensing devices. Third, for the detection of anions and cations, the sensitivity and selectivity depending on the binding affinity between inorganic ions and MOFs should be improved. Although highly selective sensors have been proposed by careful selection and design, the ability to specifically differentiate analytes with their similar structures is still weak. Finally, the fluorescence quenching mechanism for some MOFs should be systematically investigated and the quenching efficiency for biosensing should be further improved in MOFs-based biosensors. Additionally, the combination of MOFs with other materials and signal amplification strategies is helpful to achieve a higher sensitivity under the synergistic enhancement effect. It can be expected that MOFs-based fluorescent sensing platforms will possess broader application prospects under continuous efforts and in-depth investigation.

## Figures and Tables

**Figure 2 biosensors-12-00928-f002:**
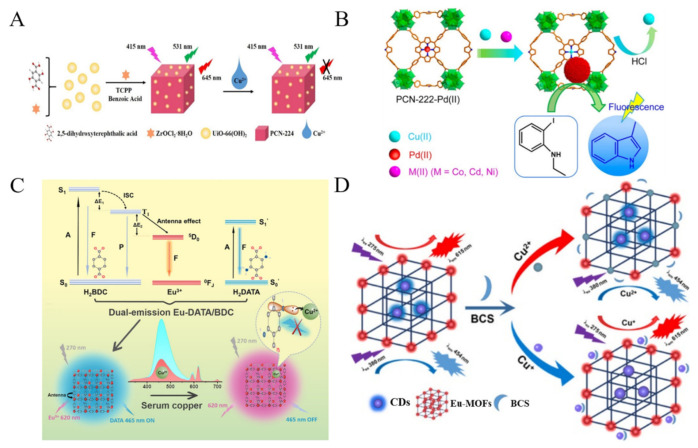
(**A**) Schematic illustration of process for encapsulating the UiO-66(OH)_2_ into the porphyrin MOFs (PCN-224) and concept for sensing Cu^2+^. Reproduced with permission [63]. Copyright 2019, American Chemical Society. (**B**) Schematic illustration of the fluorescence “turn-on” sensing of Cu^2+^ ions based on the porphyrinic MOFs-catalyzed Heck reaction. Reproduced with permission [65]. Copyright 2016, American Chemical Society. (**C**) Schematic illustration of the design principle of dual-emission Eu-DATA/BDC MOFs and the ratiometric sensing and visible detection of Cu^2+^ ions in human serum. Reproduced with permission [66]. Copyright 2022, Elsevier. (**D**) Schematic illustration of the detection mechanisms for Cu^+^ ions and Cu^2+^ ions with the dual fluorescence Eu-MOFs@CDs. Reproduced with permission [67]. Copyright 2022, American Chemical Society.

**Figure 4 biosensors-12-00928-f004:**
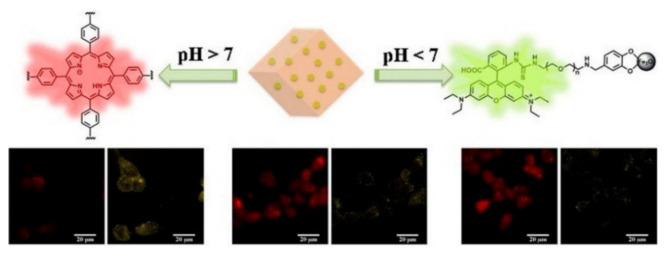
Schematic illustration of the dual-emission fluorescent MOF for monitoring the change of pH value with an excitation wavelength. Reproduced with permission [88]. Copyright 2018, American Chemical Society.

**Figure 6 biosensors-12-00928-f006:**
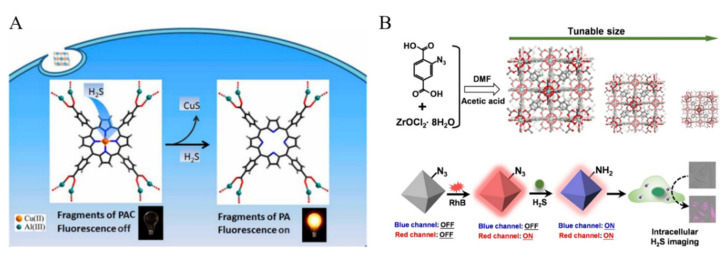
(**A**) Schematic illustration of structural fragment of nano-MOF PAC and the proposed strategy for fluorescent variation of PAC upon reactive Cu(II) ions as the H_2_S-responding site. Reproduced with permission [105]. Copyright 2014, American Chemical Society. (**B**) Schematic illustration of one-step hydrothermal synthesis of UiO-66-N_3_ in acetic acid/DMF with tunable size, the principle of H_2_S detection on the basis of RhB/UiO-66-N_3_ and their applications for intracellular H_2_S imaging. Reproduced with permission [106]. Copyright 2021, Elsevier.

**Figure 9 biosensors-12-00928-f009:**
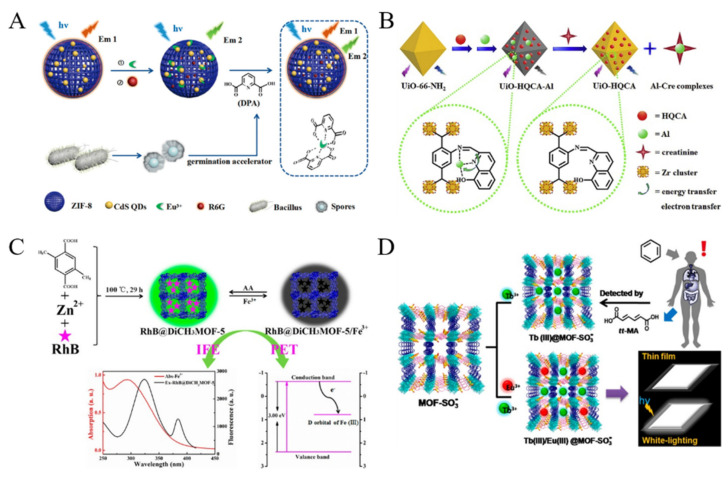
(**A**) Schematic illustration of the preparation of R6G/Eu-CdS@ZIF-8 and its application for DPA detection. Reproduced with permission [147]. Copyright 2020, American Chemical Society. (**B**) Schematic illustration of the HQCA and Al^3+^-modified MOFs-based fluorescent sensor for the detection of creatinine. Reproduced with permission [148]. Copyright 2020, Elsevier. (**C**) Schematic illustration of synthesis of fluorescent RhB@DiCH_3_MOF-5 composites and their application in Fe^3+^ and ascorbic acid detection. Reproduced with permission [149]. Copyright 2019, American Chemical Society. (**D**) Schematic illustration of the design principle of Tb(III)@MOF-SO^3−^ as a fluorescent probe for tt-MA. Reproduced with permission [150]. Copyright 2018, American Chemical Society.

**Figure 10 biosensors-12-00928-f010:**
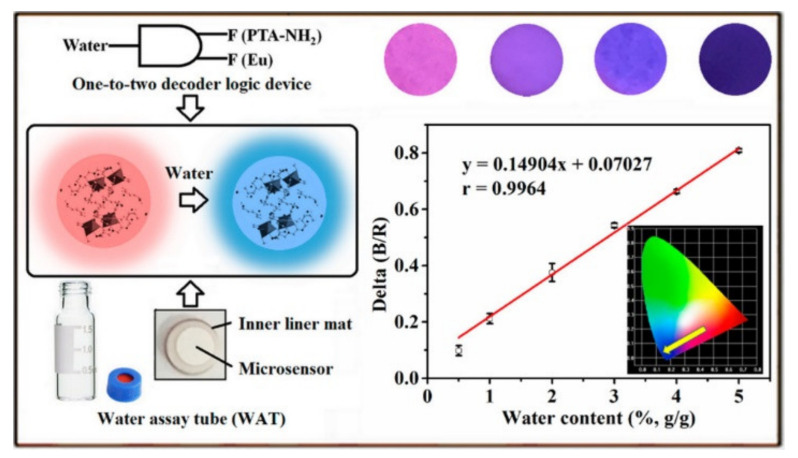
Schematic illustration of Eu-DPA/PTA-NH_2_-based portable analytical device for water analysis in solid samples. Reproduced with permission [153]. Copyright 2020, American Chemical Society.

**Figure 13 biosensors-12-00928-f013:**
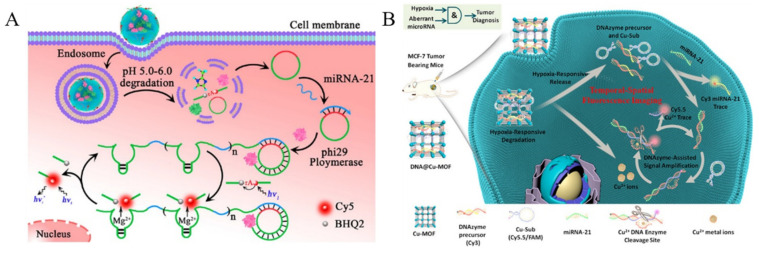
(**A**) Schematic illustration of the intracellular RCA by Using ZIF-8 NPs as a nanocarrier for codelivery of ϕ29 DNA polymerase and DNA probes for miRNA-21 imaging in vivo. Reproduced with permission [184]. Copyright 2019, American Chemical Society. (**B**) Schematic illustration of hypoxia-responsive DNA@Cu-MOF nanoprobes for aberrant miRNA and hypoxic tumor imaging using Cu^2+^ self-powered DNAzyme-assisted amplification. Reproduced with permission [185]. Copyright 2020, American Chemical Society.

**Figure 15 biosensors-12-00928-f015:**
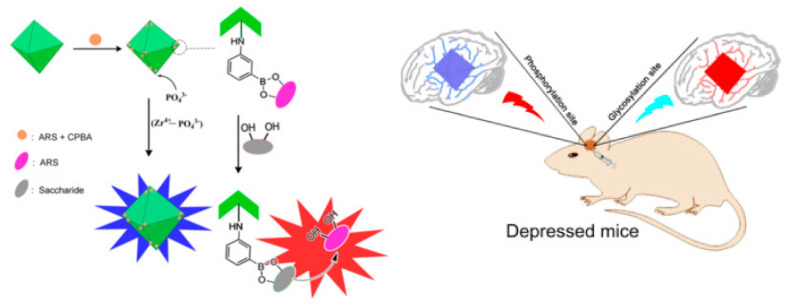
Schematic illustration of the proposed mechanism of MOFs-based detection of the levels of glycosylation and phosphorylation (left) and in-situ fluorescence imaging of the levels of glycosylation and phosphorylation in depressed mice (right). Reproduced with permission [197]. Copyright 2020, American Chemical Society.

**Figure 16 biosensors-12-00928-f016:**
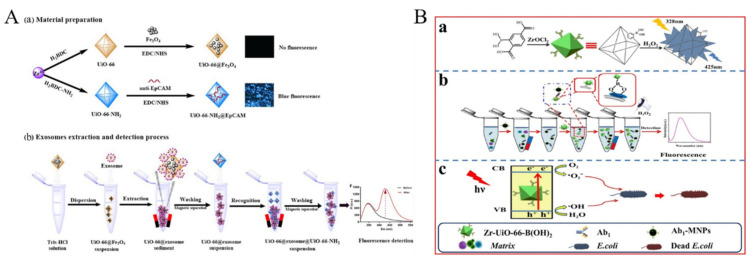
(**A**) Schematic illustration of the fluorescence detection method for cancer cell-derived exosomes based on Zr-MOFs. Reproduced with permission [199]. Copyright 2022, Elsevier. (**B**) Schematic illustration of (a) preparation of Zr-UiO-66-B(OH)_2_ and its fluorescence “turn-on” mechanism by H_2_O_2_; (b) the immunoassay process for detection of *E. coli* through fluorescence signal; (c) Mechanism underlying the photocatalytic elimination of *E. coli* by the Zr-UiO-66-B(OH)_2_. Reproduced with permission [201]. Copyright 2021, Elsevier.

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
