# Peer review of "An Overview of the Design of Metal-Organic Frameworks-Based Fluorescent Chemosensors and Biosensors"

_biosensors, 2022, doi:10.3390/bios12110928_

Round 1
Reviewer 1 Report
I recommend its publication with minor revision and re review as listed below.
1. In introduction must include about the fabrication of the sensors
2. Must include Chemosensors application compare to other sensors
3. Performances of MOFs-based fluorescent Chemosensors for the detection of inorganic ions- please include recent research works.
4. If possible include about the sensitivity of the electrodes
5. If possible include about the reproducibility of the sensors
6. If possible include about the carbon sensors
7. Review needs more comparison table like about the parameters
8. Author should maintain all the decimal places as unique. Ex: 0.124 cm2 (here 3 decimals), 0.3405 cm2 (here 4 decimals), etc.
9. Author should use all the abbreviations in the initial stage of the sentence (Ex: Author used the abbreviation of clozapine in the experimental part but not in the abstract or introduction part, etc).
10. The quality of given graphical abstract is low, author should improve.
11. Recheck all the abbreviations, equations, tables and figures (there captions).
13. Quality of all the figures is very low, author should improve.
14. The numbers in the chemical formulas should be type as subscript in all the places (Ex: HClO4 + HNO3, etc)
15. authors literature citation in biosensor field can use some fallowing reference: Journal of Analytical Science and Technology,
Author Response
We thank the reviewer for his/her positive comments: “I recommend its publication with minor revision and re review as listed below.”
Comment 1: “In introduction must include about the fabrication of the sensors.”
Response: We have added the sentences and cited the references to indicate the methods for the synthesis of MOFs and the fabrication of sensors.
Comment 2: “Must include Chemosensors application compare to other sensors.”
Response: It is a good suggestion. We have introduced the advantages of fluorescent chemosensors in contrast to other sensors in Section 3.
Comment 3: Performances of MOFs-based fluorescent Chemosensors for the detection of inorganic ions- please include recent research works.
Response: In this review, we mainly discussed the works about MOFs-based fluorescent chemosensors in recent three years. Other works about MOFs-based fluorescent chemosensors for the detection of inorganic ions have been introduced in the previous reviews, which have been cited in Part 3.1 (Refs. 58-62).
Comment 4: If possible include about the sensitivity of the electrodes.
Response: The detection linear range and detection limit of the reviewed methods have been included in the tables.
Comment 5: If possible include about the reproducibility of the sensors
Response: Most of the reported works did not investigate the reproducibility of the sensors. Thus, we only presented the linear range and detection limit of the sensors in the tables.
Comment 6: If possible include about the carbon sensors.
Response: This review did not refer to the carbon sensors.
Comment 7: Review needs more comparison table like about the parameters.
Response: The detection linear range and detection limit of the sensors have been included in the tables.
Comment 8: Author should maintain all the decimal places as unique. Ex: 0.124 cm2 (here 3 decimals), 0.3405 cm2 (here 4 decimals), etc.
Response: We did not maintain the decimal places as unique because the values in this review are directly used according to the original papers.
Comment 9: Author should use all the abbreviations in the initial stage of the sentence (Ex: Author used the abbreviation of clozapine in the experimental part but not in the abstract or introduction part, etc).
Response: We have carefully examined the abbreviations and made the modification.
Comment 10: The quality of given graphical abstract is low, author should improve.
Response: The graphical abstract for this journal is not mandatory. Thus, we did not provide the graphical abstract in the submission system.
Comment 11: Recheck all the abbreviations, equations, tables and figures (there captions).
Response: We have carefully checked the abbreviations, equations, tables and figures and made the modification.
Comment 13: Quality of all the figures is very low, author should improve.
Response: We have improve the quality of all the figures.
Comment 14: The numbers in the chemical formulas should be type as subscript in all the places (Ex: HClO4 + HNO3, etc).
Response: We have carefully checked the chemical formulas and made the modification.
Comment 15: Authors literature citation in biosensor field can use some fallowing reference: Journal of Analytical Science and Technology, 2020, 11(1), 3.
Response: We have cited the reference to support the presentation.
Author Response
We thank the reviewer for his/her positive comments: Herein the authors have collected an overview of Metal-organic frameworks for Chemo- and Biosensor applications. The review covers a wide range of metal-organic frameworks for this specific application and is suitable for publication in the journal Biosensor.
Comment: There are some recent reviews which might be included to make this review better. 1. Chem. Soc. Rev., 2021,50, 4484-4513.
- L. Liu, Y. Zhou, S. Liu, M. Xu, ChemElectroChem 2018, 5, 6.
- Nano-Micro Lett. 10, 64 (2018)
- Sensors 2017, 17, 1108; doi:10.3390/s17051108
- Cell Mol Bioeng. 2021 Dec; 14(6): 535–553.
Response: We have cited the references to support the presentation.
Reviewer 3 Report
The review article by Gao et al. provides brief review on the designing of various MOFs for chemo- and biosensing applications. The topic of research is interesting because MOFs are pioneering class of porous materials widely used in the area of separation and detection. This review provides a list of useful references with meaningful discussion. It is a really great effort to put all these molecular systems together which will be of great help to the readers interested in the topic. Therefore, considering importance of the selected topic and systematic compilation of research, I recommend this review for publication in Biosensors. My few suggestions for further improvement are as follows-
1. In the Introduction Section, authors should clearly indicate what new knowledge is gained from this review in contrast to previous reviews published in the same area. Moreover, authors are also encouraged to cite the following missing review papers related to the topic (Coordination Chemistry Reviews 2019, 401, 213065, Coordination Chemistry Reviews 2022, 463, 214539, Chemosensors 2022, 10, 412).
2. It is admirable that authors have briefly discussed the most common fluorescence sensing mechanisms such as FRET, PET, etc. It would be really convenient to the readers if the same mechanisms are described graphically by designing a figure or taking it from some literature.
Author Response
We thank the reviewer for his/her positive comments: The review article by Gao et al. provides brief review on the designing of various MOFs for chemo- and biosensing applications. The topic of research is interesting because MOFs are pioneering class of porous materials widely used in the area of separation and detection. This review provides a list of useful references with meaningful discussion. It is a really great effort to put all these molecular systems together which will be of great help to the readers interested in the topic. Therefore, considering importance of the selected topic and systematic compilation of research, I recommend this review for publication in Biosensors. My few suggestions for further improvement are as follows:
Comment 1: In the Introduction Section, authors should clearly indicate what new knowledge is gained from this review in contrast to previous reviews published in the same area. Moreover, authors are also encouraged to cite the following missing review papers related to the topic (Coordination Chemistry Reviews 2019, 401, 213065, Coordination Chemistry Reviews 2022, 463, 214539, Chemosensors 2022, 10, 412).
Response: We have cited the references to support the presentation.
Comment 2: It is admirable that authors have briefly discussed the most common fluorescence sensing mechanisms such as FRET, PET, etc. It would be really convenient to the readers if the same mechanisms are described graphically by designing a figure or taking it from some literature.
Response: It is a good suggestion. We have added Figure 1 to indicate to the mechanisms.